organic chemistry/green chemistry/biochemistry

sono-synthesis, *bis*-quinazolin-4-ones, anti-cancer agents, deep eutectic solvent

**Author for correspondence:**
Wael Abdelgayed Ahmed Arafa
e-mail: waa00@fayoum.edu.eg

This article has been edited by the Royal Society of Chemistry, including the commissioning, peer review process and editorial aspects up to the point of acceptance.

# Deep eutectic solvent for an expeditious sono-synthesis of novel series of *bis*-quinazolin-4-one derivatives as potential anti-cancer agents

## Wael Abdelgayed Ahmed Arafa[1,2]

[1]Chemistry Department, College of Science, Jouf University, PO Box 2014, Sakaka, Aljouf, Kingdom of Saudi Arabia
[2]Chemistry Department, Faculty of Science, Fayoum University, PO Box 63514, Fayoum City, Egypt

WAAA, 0000-0002-9288-4143

To produce a new category of anti-cancer compounds, a facile and environmentally sustainable method for preparing diversified *bis*-quinazolinones was demonstrated using recyclable deep eutectic solvent (DES) under ultrasonic irradiation. The reactions were performed smoothly with a wide scope of substrates affording the desired derivatives in good-to-excellent yields under an atom-economical pathway. Particularly, halogen substituents that are amenable for further synthetic elaborations are well tolerated. Furthermore, the 'greenness' of the protocol was assessed within the scope of several green metrics and found to display an excellent score in the specified parameters. Cytotoxic activity of all novel *bis*-quinazolinones was investigated utilizing two cancer cell lines: breast (MCF-7) and lung (A549) cell lines and their $IC_{50}$ values were determined. Most of the prepared derivatives displayed fascinating inhibitory activity with $IC_{50}$ values in a low micromolar range. Remarkably, the derivative **7e** [3,3'-(sulfonyl*bis*(4,1-phenylene))*bis*(2-methyl-6-nitroquinazolin-4(3*H*)-one)] showed superior potency against MCF-7 and A549 cancer cell lines, with $IC_{50}$ values of 1.26 μM and 2.75 μM, respectively. Moreover, this derivative was found to have low toxicity to the normal breast cell line (MCF-10A) and could serve as a promising lead candidate for further development.

## 1. Introduction

As an important category of nitrogen-containing heterocycles, quinazolines are prevalent in many naturally occurring alkaloids

as well as marketed drugs [1–4]. Some quinazolines which are well known as drugs, such as mecloqualone (Casfen), mebroqualone and methaqualone (Quaalude) possessed anxiolytic, calmative and hypnotic properties and are used for treating insomnia [5,6]. Presently, the quinazoline moiety is recognized to have a broad scope of beneficial biological activities, for instance, anti-cancer, anti-inflammatory, anti-tumour, protein kinase inhibitor, anti-microbial cholinesterase inhibitor, antifolate, antiviral and are the essential moiety of HIV reverse transcriptase inhibitors [7–11]. Proportional with the importance of this heterocyclic motif, considerable synthetic protocols have been reported to develop access to quinazolines with diversified substituents [12–15]. Nevertheless, most of them are restricted with a multistep procedure and low atom-economy to prepare quinazoline compounds. Also, in several cases, the utilization of metal–catalyst may enhance the reaction efficiency; metal contaminations are still an issue, essentially when the outcomes are manufactured for the consumption of humans [16–19]. Latterly, many endeavours have been reported to demonstrate easy-to-handle, cost-effective and eco-friendly protocols for the preparation of quinazolinones [20–22]. Notwithstanding, to the best of our knowledge there have been restricted articles to date on the assembly of *bis*-quinazolinones [23,24]. Subsequently, the development of an effective and greener methodology for the construction of *bis*-quinazolinones is extremely in demand. Recently, more attempts have been made for designing eco-friendly solvents that could be reusable and/or smoothly biodegradable. For example, green solvents such as supercritical liquids [25], water [26], polyethylene glycol [27] and ionic liquids [28–30] have emerged to supersede numerous organic solvents. Nevertheless, in some cases, the utilization of these solvents is limited because of their poor stability and solubility of organic substances. Lately, deep eutectic solvents (DESs), the environmentally benign solvent systems, have attracted considerable attention [31,32]. DESs possess several noteworthy merits when compared with conventional solvents, comprising a broad range of liquid temperatures, renewability, water tolerance, low toxicity, biodegradability, low volatility and high solvation capacity for an extensive number of organic substances [33–35]. DESs are easily prepared from mixing an organic hydrogen bond donor (e.g. tartaric acid) and a hydrogen bond acceptor (e.g. choline chloride) [31,32]. The obtained mixture possesses high stability and remains in a high-entropy liquid state even at low temperatures; this may be attributed to the newly formed hydrogen-bond network between its components [31,32,36,37]. Numerous reports have demonstrated their utilization in several organic conversions [38–40]. Because of these special advantages, DESs opened novel perspectives to prepare new substances. The use of ultrasonic irradiation as the green energy source is of a significant utility in the field of pharmaceutical and green chemistry. The impacts of sonication on organic synthesis are imputed to cavitations, which generate extremely high local pressure and temperature inside the formed bubbles. When these bubbles collapse, a sufficient amount of energy was generated for performing the chemical reaction and allowing the process to occur easily with a high yield in a very short time [41,42]. In spite of that, to the best of our knowledge there have been no reports to date on the preparation of *bis*-quinazolinones using deep eutectic solvents under ultrasonic irradiation. In continuation to our ongoing efforts to develop greener synthetic pathways for organic conversions [43–46], herein I wish to report a DES-mediated protocol for greener and atom-economical preparation of novel series of *bis*-quinazolinones by using commercially available and inexpensive materials under appropriate catalyst-free and ultrasonic conditions.

## 2. Results and discussion

At the outset, the present study was commenced with optimization of the reaction conditions using 2-methyl-4*H*-benzo[*d*][1,3]oxazin-4-one (**1a**) and *trans*-cyclohexane-1,4-diamine (**2a**) as model substrates (table 1). The initial endeavour utilizing conventional refluxing conditions in $CH_3CN$ and in the presence of $K_2CO_3$ as the basic catalyst [23,24] afforded the hitherto unreported 3,3'-((1*R*,4*R*)-cyclohexane-1,4-diyl)*bis*(2-methylquinazolin-4(3*H*)-one) (**3a**) but in a poor yield; instead, a considerable amount of starting materials remain unchanged (table 1, entry 1). Next, on performing the above reaction under ultrasonic irradiation (60 W) at 50°C for 20 min, furnished 40% of the product (table 1, entry 2). Anticipating further improvement in the yield, a series of base and acid catalysts, such as $Cs_2CO_3$, AcOH/NaOAc and *p*-toluenesulfonic acid (table 1, entries 3–5), were screened under ultrasonic irradiation (60 W); of these, *p*-TSA was found to be the most effective catalyst for this conversion (table 1, entry 5). Furthermore, a moderate yield was obtained under catalyst-free and ultrasound conditions, which suggest an apparent over-activity by the used catalyst (table 1, entry 6). Such an observation is also confirmed by literature studies, which proposed that ultrasonic irradiation may replace a catalyst under definite conditions [47,48]. In order to acquire more acceptable results, other reaction variables were

**Table 1.** Optimization of the reaction conditions for the preparation of compound **3a**.

| entry | catalyst | solvent | temp. (°C)/method | time (min) | yield (%)[a] |
|---|---|---|---|---|---|
| 1 | $K_2CO_3$ | $CH_3CN$ | Reflux[b] | 300 | 25 |
| 2 | $K_2CO_3$ | $CH_3CN$ | 50/US[c] | 20 | 40 |
| 3 | $Cs_2CO_3$ | $CH_3CN$ | 50/US | 20 | 43 |
| 4 | NaOAc | AcOH | 50/US | 20 | 45 |
| 5 | *p*-TSA | $CH_3CN$ | 50/US | 20 | 57 |
| 6 | — | $CH_3CN$ | 50/US | 20 | 65 |
| 7 | — | $H_2O$ | 50/US | 20 | 70 |
| 8 | — | $H_2O$ | 95/US | 35 | 77 |
| 9 | — | EtOH | 50/US | 20 | 68 |
| 10 | — | *i*-PrOH | 50/US | 20 | 57 |
| 11 | — | toluene | 50/US | 20 | 38 |
| 12 | — | THF | 50/US | 20 | 15 |
| 13 | — | sucrose–choline chloride (1 : 1) | 80/US | 20 | 17 |
| 14 | — | lactic acid–choline chloride (1 : 2) | 80/US | 20 | 73 |
| 15 | — | oxalic acid–choline chloride (1 : 1) | 80/US | 20 | 75 |
| 16 | — | L-(+)-tartaric acid–choline chloride (1 : 2) | 85/US | 20 | 89 |
| 17 | — | L-(+)-tartaric acid–choline chloride (1 : 2) | 90/US | 15 | 99 |
| 18 | — | L-(+)-tartaric acid–choline chloride (1 : 2) | 90/US | 10 | 91 |

[a]Isolated yield.
[b]The reaction was carried out under reflux conditions.
[c]The reactions were carried out under ultrasonic irradiation (US) conditions.

assessed. First, the role of additional solvents was investigated and found that the yield of **3a** was improved to be 70% when performing water as a solvent at 50°C (table 1, entry 7), and slightly increased to be 77% in a longer reaction time at 95°C (with table 1, entry 8). In addition, ethanol furnished a moderate yield at 50°C (table 1, entry 9), while other solvents such as *i*-PrOH and toluene were substantially less efficient and THF was ineffectual (table 1, entries 10–12). To evaluate the current protocol from the greener edge, deep eutectic solvents (DESs) were used as the reaction medium for this conversion. To acquire derivative **3a** (table 1), we turned our venture by using certain types of DESs with the model substrates **1a** (2.0 equiv.) and **2a** (1.0 equiv). As displayed in table 1, only a small amount of the required product (**3a**) was obtained when the reaction proceeded in sucrose–choline chloride mixture (1 : 1) at its melting temperature (table 1, entry 13). Considering that the model reaction works well under acidic conditions, three acidic DESs were examined with a view to acquiring more satisfactory results. Interestingly, a considerable improvement was obtained when DESs such as lactic acid–choline chloride (1 : 2), oxalic acid–choline chloride (1 : 1) and L-(+)-tartaric acid–choline chloride (1 : 2) were used as the reaction solvent (table 1, entries 14–16). This screening disclosed that the utilization of L-(+)-tartaric acid–choline chloride (1 : 2) melt afforded the superior result (table 1, entry 16). In a quest to further improve the reaction yield, the model substrates were sonicated at two operating temperatures, i.e. 85 and 90°C and the required product (**3a**) was obtained in 89% and 99% yields, respectively (table 1, entries 16 and 17). Thus, the optimal temperature was established to be 90°C, which achieved the best yield of the desirable product. Decreasing the sonication time to 10 min diminished the isolated yield 91% (table 1, entry 18).

After optimization of the current protocol, the impact of several ultrasonic irradiation powers on the yield and rate of model reaction were also studied (table 2). The results display that the power of

**Scheme 1.** Reaction of derivatives **2a–c** under the standard conditions.

**Table 2.** The impact of ultrasonic irradiation power on the synthesis of **3a**.

| entry | power (W) | time (min) | yield (%)[a] |
|---|---|---|---|
| 1 | 30 | 60 | 76 |
| 2 | 40 | 50 | 80 |
| 3 | 50 | 30 | 82 |
| 4 | 60 | 15 | 99 |
| 5 | 70 | 15 | 99 |
| 6 | silent (reflux)[b] | 320 | 78 |

[a]Isolated yield.
[b]The reaction was carried out under reflux conditions.

sonication has a considerable impact on the reaction system. Thus, the sonication of the model substrates in L-(+)-tartaric acid–choline chloride (1 : 2) at 30 W afforded the desired product (**3a**) in a moderate yield throughout 60 min (table 2, entry 1). While, both reaction rate and yield were improved by increasing the sonication power to 60 W (table 2, entries 2–4). Moreover, on sonication of model reactants at 70 W, neither the reaction rate nor the reaction yield was improved (table 2, entry 5). Noteworthy, on carrying out the model reaction under reflux conditions, the reaction effectiveness is obviously diminished (table 2, entry 6). The utilization of ultrasound technique reduces the reaction duration and enhances the reaction yield possibly through increasing the energy and collision speed of the reactants. Consequently, sonication (60 W) of **1a** (2 mmol) and **2a** (1 mmol) in L-(+)-tartaric acid–choline chloride (1 : 2) at 90°C for 15 min was established as the optimum conditions, affording the desired product (**3a**) in 99% yield (table 1, entry 17).

After achieving an optimized strategy for this new reaction, the substrates' versatility and scope with regard to diamines and benzoxazin-4-ones were then investigated for the assembly of various *bis*-quinazolinones (schemes 1 and 2). First, the electronic impacts of R groups on 2-methyl-4*H*-benzo[*d*][1,3]oxazin-4-ones (**1a–e**) were studied. Generally, the aryl motif linked to an electron donating substituent (e.g. –CH₃) afforded better yields than that linked to electron withdrawing substituents (e.g. –Cl, –F and –NO₂).

Furthermore, diverse diamines, comprising aromatic and aliphatic diamines, were well tolerated. Interestingly, all the reactions proceeded smoothly to provide the corresponding *bis*-quinazolinones depending on the reaction time. *trans*-Cyclohexane-1,4-diamine (**2a**) afforded the required products (**3a–e**)

**Scheme 2.** Reaction of derivatives **2d** and **2e** under the standard conditions.

**Scheme 3.** Reaction of derivatives **2f** and **2g** under the standard conditions.

**Scheme 4.** Calculated green metrics for the scaled-up preparation of 3,3′-((1R,4R)-cyclohexane-1,4-diyl)bis(2-methylquinazolin-4(3H)-one) **3a**.

in short reaction time with high yields. Also, p-xylylenediamine (**2b**) and ethylene diamine (**2c**) could not reduce the reaction rate remarkably. Furthermore, p-phenylene diamine (**2d**) and dapsone (**2e**) produced the corresponding derivatives in moderate-to-high yields but in longer reaction time (scheme 2). Owing to the fact that heterocycles bearing halogens are significant building blocks in the assembly of plentiful pharmaceuticals and natural products [49,50], an assortment of halo-substituted bis-quinazolinones, e.g. chloro **3c** (96%), **4c** (95%), **5c** (95%), **6c** (95%), **7c** (97%) and fluoro **3d** (96%), **4d** (92%), **5d** (94%), **6d** (97%), **7d** (93%), were successfully prepared.

The steric impact of the utilized diamines had an obvious effect on the effectiveness of these reactions. For instance, when the present method was put in an application for o-phenylene diamine (**2f**), however, bis-quinazolinone derivative (**9**, 67%) was isolated along with the mono-quinazolinone derivative [23,24] (**8**, 33%). Whereas, by increasing the reaction time to 25 min and molar ratio of **1a** to 2.5 mmol, the reaction yield was improved to 88% with the complete consumption of the mono-quinazolinone derivative **8** (monitored by TLC). The comparable result was acquired when m-phenylene diamine (**2 g**) was utilized as the substrate; the reaction afforded a complex mixture of compounds **10** [51] and **11** under optimum conditions. By increasing both time (30 min) and molar ratio of **1a** (2.5 mmol), compound **10** disappeared and the reaction afforded derivative **11** in 92% yield (scheme 3). The structures of all the novel bis-quinazolinones were interpreted on the basis of spectral analyses (IR, NMR spectroscopy and mass spectrometry).

In order to further display the practicality of the above-mentioned investigations, the current protocol was extended for large-scale preparation of bis-quinazolin-4(3H)-one (**3a**) (scheme 4). As outlined in scheme 4, the reaction of trans-cyclohexane-1,4-diamine (**2a**, 1.14 g, 10.0 mmol) with benzoxazine (**1a**, 3.22 g, 20.0 mmol), using L-(+)-tartaric acid–choline chloride (1:2) as the solvent, was scaled up.

**Scheme 5.** A plausible mechanism for the preparation of *bis*-quinazolin-4-ones **3 – 11**.

Under the optimal conditions, the required product **3a** was obtained in 98% yield by crystallization from dioxane. Lastly, to assess the current protocol on the 'greenness' scale, green metrics [52,53] such as E-factor (EF), atom economy (AE), reaction mass efficiency (RME), process mass intensity (PMI), yield economy (YE) and carbon efficiency (CE) were studied. As summarized in scheme 4, the current protocol achieved a good combination of AE (91.74%), CE (100%), EF (0.02), RME (98.67%), PMI (1.11) and YE (6.53%), which makes it an ideal sustainable and green process. The results might be beneficial from environmental and industrial perspectives.

According to the aforementioned experiments and the reported literature [54], a proposed mechanism for the reaction between diamines (**2a–g**) and benzoxazines (**1a–e**) was established as presented in scheme 5. The first step could be the ring opening via nucleophilic attack of amine (NH2) to carbonyl carbon under the presence of L-(+)-tartaric acid–choline chloride. Next, the formed intermediate underwent intramolecular nucleophilic attack of the nitrogen (−NH−) of the amide group to carbonyl carbon. Finally, the required product was formed through dehydration [54]. From the proposed mechanism, it is obviously concluded that, L-(+)-tartaric acid–choline chloride performs a dual function; as solvent and as catalyst.

Recyclability of the solvent is a needful merit for industrial processes. The recyclability of L-(+)-tartaric acid–choline chloride in this investigation was proceeded in the model substrates; 2-methyl-4*H*-benzo[*d*][1,3]oxazin-4-one (**1a**) and *trans*-cyclohexane-1,4-diamine (**2a**) under optimal conditions (figure 1). After consumption of reactants, the reaction contents were cooled to ambient temperature and diluted with ethanol. The desired product was filtered and washed with ethanol. After removal of ethanol under vacuum, DES was recovered and re-used for the next cycles. The recyclability proceeded five times without considerably lowering the reaction efficiency (figure 1).

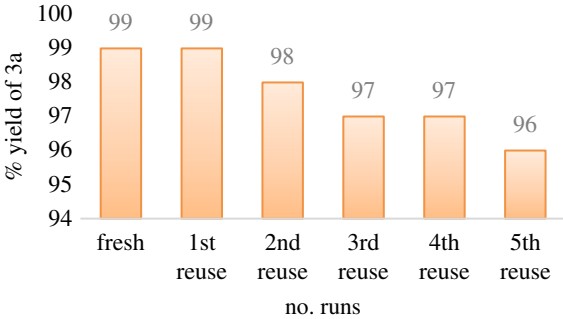

**Figure 1.** Recyclability of DES for the synthesis of **3a**.

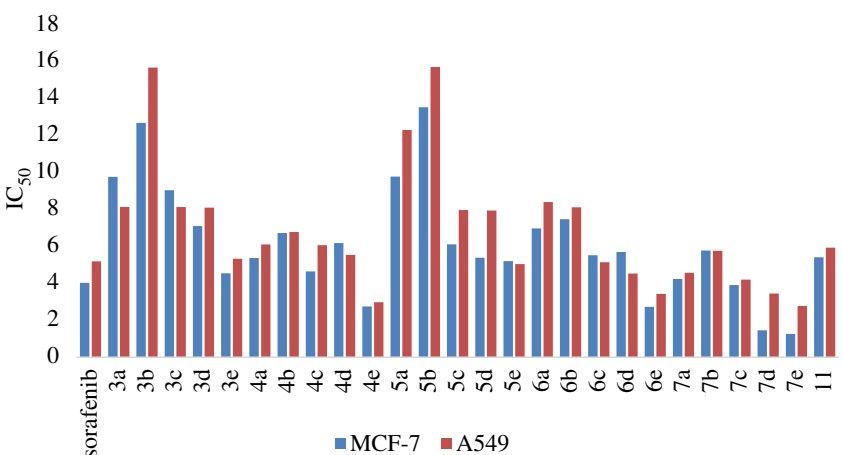

**Figure 2.** *In vitro* studies of the synthesized compounds (**3–7** and **11**) against human cancer cell lines (MCF-7 and A549).

# 3. Assessment of *in vitro* human cancer cell lines growth inhibition and the structure and activity relationships (SARs)

The unprecedented derivatives (**3–7** and **11**) were tested for their *in vitro* cytotoxicity against human MCF-7 (breast cancer), A549 (lung cancer) cancer cell lines and one type of normal cell line MCF10A (normal breast cell line) utilizing the standard MTT (3,4-dimethylthiazol-2-yl)-2,5-diphenyltetrazolium bromide assay according to Mosmann's methodology [55] (figure 2), whereas sorafenib was performed as a standard compound. Two concentrations (10 and 30 µM in triplicate) of the derivatives were utilized and after 48 h of drug treatment the results were analyzed. The obtained results were tabulated as mean ± s.e.m. With standard errors below 10%, the reproducibility between replicate wells is within the acceptable values. Concentrations less than 10 µM have negligible impacts on the cytotoxicity. The investigation of the SARs displayed that the substituent R groups in the quinazolinone moieties have a significant role in their biological properties. The results summed up in electronic supplementary material, S59, table S3 indicate that all of the tested derivatives have good-to-excellent cytotoxic activities, with the $IC_{50}$ values at the micromolar range (figure 2). Regarding the MCF-7 breast cancer cell line, compounds **7a–e** elicited potent anti-breast cancer activity ($IC_{50} = 1.26$– 5.78 µM), compared with sorafenib ($IC_{50} = 4.03$ µM). This reactivity could be related to the high potency of both dapsone and quinazolinone moieties, when compared with the other derivatives. Noteworthy, compounds **4a–e** comprising *p*-xylylene moiety resulted in good-to-excellent activities with $IC_{50}$ of 2.75–6.73 µM. *Bis*-quinazolin-4-one derivatives **5a–e** ($IC_{50} = 5.21$–13.55 µM) were less active than the cyclohexane derivatives **3a–e** ($IC_{50} = 4.55$–12.70 µM). Whereas, 1,4-phenylene derivatives **6a–e** (2.73–7.48 µM) were slightly more active than cyclohexane derivatives **3a–e**. Similarly, derivative **11** possessed good activity with an $IC_{50}$ value of 5.42 µM. Moreover, substitution with electron withdrawing groups (Cl, F and $NO_2$) displayed good activity, while electron donating group ($CH_3$) reduced the activity. A comparison of the substituents on the quinazolinone ring

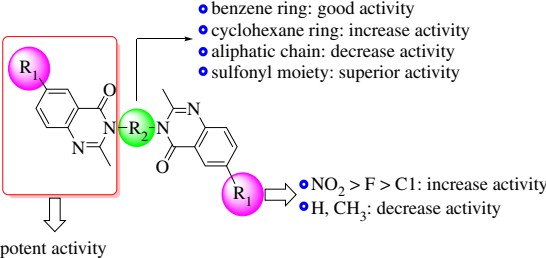

**Figure 3.** Illustration of the structural activity relationship of the designed derivatives.

indicated that nitro groups significantly enhanced the cytotoxic activity and determined the order of potency. For instance, derivative **7e** with $NO_2$-substitution at the quinazolinone moiety was found to be the most potent in the series with $IC_{50}$ value of 1.26 μM against MCF-7, being approximately thrice as potent as sorafenib. Likewise, derivatives **7c** and **7d** possessed a more important activity than unsubstituted quinazolinone moiety (**7a**) and methyl substituted derivative (**7b**).

Concerning the A549 lung cancer cell line (electronic supplementary material, table S3), the alteration of substituents on 6-position of the quinazolinone ring could also influence the activities of these compounds. As summarized in figure 2, the potency of substituents was ordered as $NO_2 > F > Cl > H > CH_3$, indicating that those with the electron withdrawing group are more active than those with an electron donating substitute. Ten derivatives, **7a–e**, **6c–e**, **5e** and **4e**, possessed promising cytotoxic activities as indicated from their $IC_{50}$ values (2.78–5.77 μM) compared with sorafenib ($IC_{50} =$ 5.20 μM). Among them, derivative **7e**, 3,3′-(sulfonyl*bis*(4,1-phenylene))*bis*(2-methyl-6-nitroquinazolin-4(3*H*)-one), showed the best activity with $IC_{50}$ value of 2.75 μM against A549, being approximately twice as potent as sorafenib. The series of derivatives **6** and compound **11**, with aryl motif, also afforded potent activities and displayed similar tendencies in the relationship between the structure and activity. Nevertheless, the series of compounds **3** and **5** afforded slightly weaker activities (figure 2). Figure 3 clarifies the structural activity relationship of the designed derivatives.

In order to decide whether the cytotoxic properties of the investigated derivatives were selective for cancerous cells in contrast to non-cancerous cells, the most active derivatives, viz. **4e**, **6e**, **7d** and **7e**, were subjected to *in vitro* assessment against the normal breast cell line (MCF-10A). The results of the non-malignant cell (MCF-10A) demonstrated that derivatives **7e** ($IC_{50} = 7.17 \pm 0.76$ μM), **7d** ($IC_{50} =$ $5.98 \pm 1.86$ μM), **6e** ($IC_{50} = 5.65 \pm 2.34$ μM) and **4e** ($IC_{50} = 5.43 \pm 2.46$ μM) exhibited promising results in comparison with sorafenib ($IC_{50} = 5.98 \pm 0.49$ μM) when tested under the same conditions. From the obtained results, we can conclude that these derivatives might be further used as promising anti-cancer agents.

In the present investigation, incorporation of two symmetrical pharmacophores into a single molecule resulted in designed multiple compounds possessing fascinating inhibition activities against clinically relevant objectives that are not obtainable by their monomeric analogues.

# 4. Conclusion

A straightforward, green and effective protocol for synthesizing new series of *bis*-quinazolin-4-ones from easily obtainable precursors under minutes of ultrasonic irradiation was demonstrated on a gram-scale. Noteworthy, the L-(+)-tartaric acid–choline chloride deep eutectic solvent displayed to be appropriate for these transformations as indicated by its recyclability, eco-friendly and inexpensive preparation, and simplicity of operation. Also, the sonochemical methodology performs well for both activating and deactivating starting substrates. The reaction yields are excellent and the protocol is devoid of column purification and any tedious work-up. An assortment of halo-substituted products was successfully synthesized via the current strategy that provides possibility for further cross-coupling reactions. As well, the protocol scored well in several green metrics and consequently may be of sustainable and practical benefit in the future. All the newly prepared derivatives possessed potential cytotoxic activities, with the strength of the impact depending on the substituent on the quinazolinone moiety. Among them, derivatives **7e**, 3,3′-(sulfonyl*bis*(4,1-phenylene))*bis*(2-methyl-6-nitroquinazolin-4(3*H*)-one), exhibited the most potent cytotoxic activities and possessed the ability to kill malignant cells more effectively than non-malignant cells.

# 5. Experiment

## 5.1. General information

$^1$H, $^{19}$F and $^{13}$C NMR spectra were recorded on Bruker Ultra Shield spectrometer at 400, 376 and 100 MHz, respectively. Mass spectra were determined on the GC-MS (QP/000 EX) Shimadzu spectrometer at an ionizing voltage of 70 eV. IR spectra were recorded on the Perkin-Elmer Spectrum One spectrometer using KBr pellets. Analytical thin layer chromatography (TLC) was employed on a silica gel plate (Merck® 60F254). Sonication was performed in a SY5200DH-T ultrasound cleaner. Melting points were measured on Electrothermal IA9100 melting point apparatus (UK) using open capillary tubes and are uncorrected. All commercial reagents were used as received without any purification. Cytotoxic activities were performed by the Microanalysis Center, Faculty of Science, Cairo University. Derivatives **1a−e** were synthesized by the sonication of anthranilic acid derivatives and acetic anhydride at 40°C for 10 min.

## 5.2. Synthesis of DESs

The DESs were synthesized by mixing choline chloride with several hydrogen bond donors such as sucrose, lactic acid, oxalic acid and tartaric acid, according to the molar ratio mentioned in table 1. The two components of each mixture were heated at 85°C under sonication until a clear liquid was obtained, which was directly used for *bis*-quinazolin-4(3H)-ones synthesis.

## 5.3. Typical procedure for the preparation of *bis*-quinazolin-4(3H)-ones 3−11

2-Methyl-4H-benzo[d][1,3]oxazin-4-ones (**1a−e**) (2 mmol), diamines (**2a−g**) (1 mmol) and L-(+)-tartaric acid−choline chloride (1 : 2) based DES (4 ml) were added to a 25 ml round bottom flask and the reaction mixture was sonicated (60 W) for 15 min (20−30 min in the case of using **2d−g**) at 90°C. The progress of the reaction was monitored by TLC. After completion of the reaction, the mixture was cooled to room temperature and diluted with ethanol. The desired product was obtained by filtration and recrystallized from dioxane containing a few drops of DMF. The filtrate which mainly contained DES was dried under vacuum to get DES. DES so recovered was utilized for the next experiments. The recyclability study was performed five times and displayed no significant drop in the reaction yields.

### 5.3.1. 3,3′-((1R,4R)-Cyclohexane-1,4-diyl)*bis*(2-methylquinazolin-4(3H)-one) 3a

White crystals, yield 99%, m.p. 273−275°C; IR (KBr): $\nu$/cm$^{-1}$ 1672 (C=O), 1620, 1595 (C=N, C=C); $^1$H NMR (400 MHz, DMSO-$d_6$): δ 8.37−8.35 (d, J = 5.5 Hz, 2H, Ar-H), 7.87−7.85 (d, J = 5.6 Hz, 2H, Ar-H), 7.44−7.31 (m, 4H, Ar-H), 3.74−3.61 (m, 2H, Cyclohexane-H), 2.30 (s, 6H, CH$_3$), 1.65 (br, 4H, Cyclohexane-H), 1.44−1.31 (m, 4H, Cyclohexane-H); $^{13}$C NMR (100 MHz, DMSO-$d_6$): δ 170.1 (C=O), 147.6 (C=N), 146.0, 134.5, 128.1, 127.5, 127.1, 126.7 (Ar-C), 52.5 (Cyclohexane-C), 31.1 (Cyclohexane-C), 20.3 ppm (CH$_3$); MS (EI): m/z (%) 401 (M$^+$+1, 4.6), 400 (M$^+$, 100); Anal. Calcd for C$_{24}$H$_{24}$N$_4$O$_2$: C, 71.98; H, 6.04; N, 13.99%; Found: C, 71.93; H, 5.99; N, 13.90%.

### 5.3.2. 3,3′-((1R,4R)-Cyclohexane-1,4-diyl)*bis*(2,6-dimethylquinazolin-4(3H)-one) 3b

White crystals, yield 99%, m.p. 280−281°C; IR (KBr): $\nu$/cm$^{-1}$ 1678 (C=O), 1613, 1590 (C=N, C=C); $^1$H NMR (400 MHz, CDCl$_3$): δ 8.25 (s, 2H, Ar-H), 7.90−7.87 (d, J = 7.5 Hz, 2H, Ar-H), 7.59−7.55 (d, J = 7.8 Hz, 2H, Ar-H), 3.69 (br, 2H, Cyclohexane-H), 2.75 (s, 6H, CH$_3$), 2.30 (s, 6H, CH$_3$), 1.72 (br, 4H, Cyclohexane-H), 1.20 (m, 4H, Cyclohexane-H); $^{13}$C NMR (100 MHz, CDCl$_3$): δ 170.8 (C=O), 147.2 (C=N), 145.6, 138.7, 133.5, 128.0, 126.5, 122.1 (Ar-C), 52.4 (Cyclohexane-C), 31.0 (Cyclohexane-C), 20.8 ppm (CH$_3$), 20.2 ppm (CH$_3$); MS (EI): m/z (%) 429 (M$^+$+1, 7.3), 428 (M$^+$, 100); Anal. Calcd for C$_{26}$H$_{28}$N$_4$O$_2$: C, 72.87; H, 6.59; N, 13.07%; Found: C, 72.86; H, 6.62; N, 13.01%.

### 5.3.3. 3,3′-((1R,4R)-Cyclohexane-1,4-diyl)*bis*(6-chloro-2-methylquinazolin-4(3H)-one) 3c

White crystals, yield 96%, m.p. 311−313°C; IR (KBr): $\nu$/cm$^{-1}$ 1670 (C=O), 1611, 1587 (C=N, C=C); $^1$H NMR (400 MHz, CDCl$_3$): δ 8.11 (s, 2H, Ar-H), 7.92−7.90 (d, J = 7.7 Hz, 2H, Ar-H), 7.65−7.64 (d, J = 5.8 Hz, 2H, Ar-H), 3.29−3.27 (m, 2H, Cyclohexane-H), 2.31 (s, 6H, CH$_3$), 1.97−1.95 (m, 4H,

Cyclohexane-H), 1.74–1.69 (m, 4H, Cyclohexane-H); $^{13}$C NMR (100 MHz, CDCl$_3$): δ 171.0 (C=O), 144.5 (C=N), 145.0, 133.8, 131.6, 128.3, 128.2, 122.6 (Ar-C), 51.7 (Cyclohexane-C), 31.3 (Cyclohexane-C), 20.4 ppm (CH$_3$); MS (EI): m/z (%) 472 (M$^+$+4, 10.7), 470 (M$^+$+2, 66.3), 468 (M$^+$, 100); Anal. Calcd for C$_{24}$H$_{22}$Cl$_2$N$_4$O$_2$: C, 61.42; H, 4.72; N, 11.94%; Found: C, 61.39; H, 4.79; N, 11.88%.

### 5.3.4. 3,3′-((1R,4R)-Cyclohexane-1,4-diyl)bis(6-fluoro-2-methylquinazolin-4(3H)-one) 3d

White crystals, yield 96%, m.p. 262–264°C; IR (KBr): ν/cm$^{-1}$ 1682 (C=O), 1616, 1597 (C=N, C=C); $^1$H NMR (400 MHz, CDCl$_3$): δ 8.12 (s, 2H, Ar-H), 7.81–7.80 (d, J = 5.5 Hz, 2H, Ar-H), 7.709–7.702 (m, 2H, Ar-H), 3.79 (br, 2H, Cyclohexane-H), 2.41 (s, 6H, CH$_3$), 1.71–1.64 (m, 4H, Cyclohexane-H), 1.33–1.26 (m, 4H, Cyclohexane-H); $^{13}$C NMR (100 MHz, CDCl$_3$): δ 171.1 (C=O), 144.5 (C=N), 159.8, 142.0, 123.8, 121.8, 118.8, 115.2 (Ar-C), 51.8 (Cyclohexane-C), 31.6 (Cyclohexane-C), 20.7 ppm (CH$_3$); $^{19}$F NMR (376 MHz, CDCl$_3$): δ -125.90; MS (EI): m/z (%) 437 (M$^+$+1, 13.8), 436 (M$^+$, 100); Anal. Calcd for C$_{24}$H$_{22}$F$_2$N$_4$O$_2$: C, 66.05; H, 5.08; N, 12.84%; Found: C, 66.11; H, 5.01; N, 12.80%.

### 5.3.5. 3,3′-((1R,4R)-Cyclohexane-1,4-diyl)bis(2-methyl-6-nitroquinazolin-4(3H)-one) 3e

Light yellow crystals, yield 96%, m.p. 301–304°C; IR (KBr): ν/cm$^{-1}$ 1675 (C=O), 1612, 1585 (C=N, C=C); $^1$H NMR (400 MHz, DMSO-d$_6$): δ 8.80 (s, 2H, Ar-H), 8.13–8.11 (d, J = 7.5 Hz, 2H, Ar-H), 7.79–7.77 (d, J = 7.8 Hz, 2H, Ar-H), 3.71 (br, 2H, Cyclohexane-H), 2.63 (s, 6H, CH$_3$), 1.97–1.95 (m, 4H, Cyclohexane-H), 1.74–1.69 (m, 4H, Cyclohexane-H); $^{13}$C NMR (100 MHz, DMSO-d$_6$): δ 170.4 (C=O), 145.8 (C=N), 151.8, 143.3, 128.4, 124.8, 119.6, 117.4 (Ar-C), 52.3 (Cyclohexane-C), 31.9 (Cyclohexane-C), 20.4 ppm (CH$_3$); MS (EI): m/z (%) 491 (M$^+$+1, 7.3), 490 (M$^+$, 100); Anal. Calcd for C$_{24}$H$_{22}$N$_6$O$_6$: C, 58.77; H, 4.52; N, 17.13%; Found: C, 58.79; H, 4.48; N, 17.09%.

### 5.3.6. 3,3′-(1,4-Phenylenebis(methylene))bis(2-methylquinazolin-4(3H)-one) 4a

White crystals, yield 98%, m.p. 277–279°C; IR (KBr): ν/cm$^{-1}$ 1688 (C=O), 1616, 1605 (C=N, C=C); $^1$H NMR (400 MHz, DMSO-d$_6$): δ 8.32–8.29 (d, J = 7.7 Hz, 2H, Ar-H), 7.91–7.90 (d, J = 5.6 Hz, 2H, Ar-H), 7.767–7.761 (m, 2H, Ar-H), 7.659–7.650 (d, J = 5.5 Hz, 2H, Ar-H), 7.30 (s, 4H, Ar-H), 4.78 (s, 4H, CH$_2$), 2.29 ppm (s, 6H, CH$_3$); $^{13}$C NMR (100 MHz, DMSO-d$_6$): δ 170.7 (C=O), 144.6 (C = N), 146.7, 134.8, 134.2, 128.2, 127.0, 126.9, 126.2, 122.5 (Ar-C), 62.5 (CH$_2$), 20.9 ppm (CH$_3$); MS (EI): m/z (%) 423 (M$^+$+1, 8.9), 422 (M$^+$, 100); Anal. Calcd for C$_{26}$H$_{22}$N$_4$O$_2$: C, 73.92; H, 5.25; N, 13.26%; Found: C, 73.89; H, 5.29; N, 13.21%.

### 5.3.7. 3,3′-(1,4-Phenylenebis(methylene))bis(2,6-dimethylquinazolin-4(3H)-one) 4b

White crystals, yield 99%, m.p. 285–287°C; IR (KBr): ν/cm$^{-1}$ 1682 (C=O), 1623, 1611 (C=N, C=C); $^1$H NMR (400 MHz, DMSO-d$_6$): δ 8.39 (s, 2H, Ar-H), 7.98–7.96 (d, J = 7.7 Hz, 2H, Ar-H), 7.41–7.29 (m, 6H, Ar-H), 4.81 (s, 4H, CH$_2$), 2.67 (s, 6H, CH$_3$), 2.29 ppm (s, 6H, CH$_3$); $^{13}$C NMR (100 MHz, DMSO-d$_6$): δ 171.2 (C=O), 145.1 (C=N), 142.9, 136.7, 134.4, 132.6, 128.0, 127.5, 126.5, 121.2 (Ar-C), 62.7 (CH$_2$), 21.8 (CH$_3$), 20.6 ppm (CH$_3$); MS (EI): m/z (%) 451 (M$^+$+1, 11.5), 450 (M$^+$, 100); Anal. Calcd for C$_{28}$H$_{26}$N$_4$O$_2$: C, 74.65; H, 5.82; N, 12.44%; Found: C, 74.61; H, 5.87; N, 12.39%.

### 5.3.8. 3,3′-(1,4-Phenylenebis(methylene))bis(6-chloro-2-methylquinazolin-4(3H)-one) 4c

Light yellow crystals, yield 95%, m.p. 327–330°C; IR (KBr): ν/cm$^{-1}$ 1675 (C = O), 1615, 1602 (C = N, C = C); $^1$H NMR (400 MHz, DMSO-d$_6$): δ 8.37 (s, 2H, Ar-H), 7.82–7.81 (d, J = 7.5 Hz, 2H, Ar-H), 7.29 (s, 4H, Ar-H), 7.13–7.06 (m, 2H, Ar-H), 4.81 (s, 4H, CH$_2$), 2.25 ppm (s, 6H, CH$_3$); $^{13}$C NMR (100 MHz, DMSO-d$_6$): δ 171.0 (C=O), 144.8 (C=N), 144.9, 133.6, 132.2, 128.2, 127.8, 127.4, 127.1, 121.2 (Ar-C), 62.8 (CH$_2$), 20.3 ppm (CH$_3$); MS (EI): m/z (%) 494 (M$^+$+4, 11.1), 492 (M$^+$+2, 65.3), 490 (M$^+$, 100); Anal. Calcd for C$_{26}$H$_{20}$Cl$_2$N$_4$O$_2$: C, 63.55; H, 4.10; N, 11.40%; Found: C, 63.59; H, 4.05; N, 11.36%.

### 5.3.9. 3,3′-(1,4-Phenylenebis(methylene))bis(6-fluoro-2-methylquinazolin-4(3H)-one) 4d

White crystals, yield 92%, m.p. 217–219°C; IR (KBr): ν/cm$^{-1}$ 1680 (C = O), 1611, 1593 (C = N, C = C); $^1$H NMR (400 MHz, DMSO-d$_6$): δ 8.01–8.00 (d, J = 5.6 Hz, 2H, Ar-H), 7.81–7.79 (m, 2H, Ar-H), 7.54–7.49 (m, 2H, Ar-H), 7.15 (s, 4H, Ar-H), 4.78 (s, 4H, CH$_2$), 2.29 ppm (s, 6H, CH$_3$); $^{13}$C NMR (100 MHz, DMSO-d$_6$): δ 171.0 (C = O), 145.4 (C = N), 160.8, 141.8, 134.6, 128.1, 127.9, 124.3, 120.2, 117.8 (Ar-C), 62.8 (CH$_2$),

20.3 ppm (CH$_3$); $^{19}$F NMR (376 MHz, DMSO-$d_6$): $\delta$ -125.91; MS (EI): $m/z$ (%) 459 (M$^+$+1, 11.8), 458 (M$^+$, 100); Anal. Calcd for C$_{26}$H$_{20}$F$_2$N$_4$O$_2$: C, 68.11; H, 4.40; N, 12.22%; Found: C, 68.07; H, 4.48; N, 12.17%.

### 5.3.10. 3,3′-(1,4-Phenylene*bis*(methylene))*bis*(2-methyl-6-nitroquinazolin-4(3*H*)-one) **4e**

Light yellow crystals, yield 93%, m.p. 350–353°C; IR (KBr): $\nu$/cm$^{-1}$ 1675 (C = O), 1601, 1593 (C = N, C = C); $^1$H NMR (400 MHz, DMSO-$d_6$): $\delta$ 8.69 (s, 2H, Ar-H), 8.36–8.35 (d, $J$ = 5.6 Hz, 2H, Ar-H), 7.87–7.85 (d, $J$ = 7.6 Hz, 2H, Ar-H), 7.29 (s, 4H, Ar-H), 4.78 (s, 4H, CH$_2$), 2.29 ppm (s, 6H, CH$_3$); $^{13}$C NMR (100 MHz, DMSO-$d_6$): $\delta$ 170.7 (C = O), 145.6 (C = N), 153.1, 142.8, 134.4, 128.4, 128.2, 123.1, 118.0, 117.4 (Ar-C), 62.5 (CH$_2$), 20.4 ppm (CH$_3$); MS (EI): $m/z$ (%) 513 (M$^+$+1, 2.6), 512 (M$^+$, 100); Anal. Calcd for C$_{26}$H$_{20}$N$_6$O$_6$: C, 60.94; H, 3.93; N, 16.40%; Found: C, 61.00; H, 3.91; N, 16.38%.

### 5.3.11. 3,3′-(Ethane-1,2-diyl)*bis*(2-methylquinazolin-4(3*H*)-one) **5a**

White crystals, yield 98%, m.p. 190–192°C; IR (KBr): $\nu$/cm$^{-1}$ 1679 (C = O), 1612, 1590 (C = N, C = C); $^1$H NMR (400 MHz, DMSO-$d_6$): $\delta$ 8.29–8.26 (d, $J$ = 7.1 Hz, 2H, Ar-H), 7.78 (s, 2H, Ar-H), 7.59–7.48 (t, $J$ = 7.4 Hz, 2H, Ar-H), 7.25–7.21 (d, $J$ = 7.1 Hz, 2H, Ar-H), 3.78 (s, 4H, CH$_2$), 2.30 ppm (s, 6H, CH$_3$); $^{13}$C NMR (100 MHz, DMSO-$d_6$): $\delta$ 171.0 (C = O), 144.9 (C = N), 143.9, 132.7, 127.4, 126.4, 126.0, 121.3 (Ar-C), 42.5 (CH$_2$), 20.4 ppm (CH$_3$); MS (EI): $m/z$ (%) 347 (M$^+$+1, 1.8), 346 (M$^+$, 100) [23,24].

### 5.3.12. 3,3′-(Ethane-1,2-diyl)*bis*(2,6-dimethylquinazolin-4(3*H*)-one) **5b**

White crystals, yield 97%, m.p. 211–212°C; IR (KBr): $\nu$/cm$^{-1}$ 1675 (C = O), 1608, 1590 (C = N, C = C); $^1$H NMR (400 MHz, CDCl$_3$): $\delta$ 8.00 (s, 2H, Ar-H), 7.80–7.79 (d, $J$ = 7.4 Hz, 2H, Ar-H), 7.50–7.49 (d, $J$ = 7.4 Hz, 2H, Ar-H), 3.80 (s, 4H, CH$_2$), 2.80 (s, 6H, CH$_3$), 2.30 ppm (s, 6H, CH$_3$); $^{13}$C NMR (100 MHz, CDCl$_3$): $\delta$ 171.2 (C = O), 144.6 (C = N), 143.3, 137.2, 134.6, 127.9, 126.2, 122.6 (Ar-C), 42.7 (CH$_2$), 21.9 (CH$_3$), 20.3 ppm (CH$_3$); MS (EI): $m/z$ (%) 375 (M$^+$+1, 3.2), 374 (M$^+$, 100); Anal. Calcd for C$_{22}$H$_{22}$N$_4$O$_2$: C, 70.57; H, 5.92; N, 14.96%; Found: C, 70.60; H, 5.89; N, 14.91%.

### 5.3.13. 3,3′-(Ethane-1,2-diyl)*bis*(6-chloro-2-methylquinazolin-4(3*H*)-one) **5c**

White crystals, yield 95%, m.p. 244–246°C; IR (KBr): $\nu$/cm$^{-1}$ 1680 (C=O), 1618, 1603 (C=N, C=C); $^1$H NMR (400 MHz, DMSO-$d_6$): $\delta$ 7.94 (s, 2H, Ar-H), 7.60–7.59 (d, $J$ = 7.5 Hz, 2H, Ar-H), 7.20–7.19 (d, $J$ = 7.4 Hz, 2H, Ar-H), 3.70 (s, 4H, CH$_2$), 2.29 ppm (s, 6H, CH$_3$); $^{13}$C NMR (100 MHz, DMSO-$d_6$): $\delta$ 170.5 (C=O), 144.9 (C=N), 133.0, 132.5, 127.6, 127.2, 122.0 (Ar-C), 42.9 (CH$_2$), 20.5 ppm (CH$_3$); MS (EI): $m/z$ (%) 418 (M$^+$+4, 11.2), 416 (M$^+$+2, 64.6), 414 (M$^+$, 100); Anal. Calcd for C$_{20}$H$_{16}$Cl$_2$N$_4$O$_2$: C, 57.85; H, 3.88; N, 13.49%; Found: C, 57.81; H, 3.91; N, 13.42%.

### 5.3.14. 3,3′-(Ethane-1,2-diyl)*bis*(6-fluoro-2-methylquinazolin-4(3*H*)-one) **5d**

White crystals, yield 94%, m.p. 219–221°C; IR (KBr): $\nu$/cm$^{-1}$ 1676 (C=O), 1609, 1601 (C=N, C=C); $^1$H NMR (400 MHz, DMSO-$d_6$): $\delta$ 7.60–7.57 (d, $J$ = 7.7 Hz, 2H, Ar-H), 7.53–7.51 (d, $J$ = 7.6 Hz, 2H, Ar-H), 7.39–7.35 (d, $J$ = 7.7 Hz, 2H, Ar-H), 3.73 (s, 4H, CH$_2$), 2.34 ppm (s, 6H, CH$_3$); $^{19}$F NMR (376 MHz, DMSO-$d_6$): -125.96; $^{13}$C NMR (100 MHz, DMSO-$d_6$): $\delta$ 170.8 (C=O), 145.5 (C=N), 160.8, 143.1, 124.5, 122.1, 121.1, 113.9 (Ar-C), 42.8 (CH$_2$), 20.5 ppm (CH$_3$); MS (EI): $m/z$ (%) 383 (M$^+$+1, 4.9), 382 (M$^+$, 100); Anal. Calcd for C$_{20}$H$_{16}$F$_2$N$_4$O$_2$: C, 62.82; H, 4.22; N, 14.65%; Found: C, 62.87; H, 4.18; N, 14.60%.

### 5.3.15. 3,3′-(Ethane-1,2-diyl)*bis*(2-methyl-6-nitroquinazolin-4(3*H*)-one) **5e**

Yellow crystals, yield 94%, m.p. 278–279°C; IR (KBr): $\nu$/cm$^{-1}$ 1683 (C=O), 1613, 1603 (C=N, C=C); $^1$H NMR (400 MHz, DMSO-$d_6$): $\delta$ 8.61 (s, 2H, Ar-H), 8.41–8.40 (d, $J$ = 7.7 Hz, 2H, Ar-H), 7.809–7.802 (d, $J$ = 7.8 Hz, 2H, Ar-H), 3.60 (s, 4H, CH$_2$), 2.29 ppm (s, 6H, CH$_3$); $^{13}$C NMR (100 MHz, DMSO-$d_6$): $\delta$ 171.3 (C=O), 144.8 (C=N), 152.6, 143.0, 128.3, 125.3, 120.0, 115.9 (Ar-C), 42.8 (CH$_2$), 20.5 ppm (CH$_3$); MS (EI): $m/z$ (%) 437 (M$^+$+1, 6.2), 436 (M$^+$, 100); Anal. Calcd for C$_{20}$H$_{16}$N$_6$O$_6$: C, 55.05; H, 3.70; N, 19.26%; Found: C, 54.98; H, 3.74; N, 19.22%.

### 5.3.16. 3,3′-(1,4-Phenylene)*bis*(2-methylquinazolin-4(3*H*)-one) **6a**

Yellow crystals, yield 98%, m.p. 178–180°C; IR (KBr): $\nu$/cm$^{-1}$ 1687 (C=O), 1604, 1598 (C=N, C=C); $^1$H NMR (400 MHz, DMSO-$d_6$): $\delta$ 8.30–8.28 (d, $J$ = 7.5 Hz, 2H, Ar-H), 8.09–8.01 (m, 2H, Ar-H), 7.78–7.74

(d, $J$ = 7.6 Hz, 2H, Ar-H), 7.61–7.54 (m, 2H, Ar-H), 7.28 (s, 4H, Ar-H), 2.29 ppm (s, 6H, C$H_3$); $^{13}$C NMR (100 MHz, DMSO-$d_6$): δ 170.1 (C=O), 145.2 (C=N), 136.8, 134.3, 130.8, 129.2, 127.3, 126.9, 122.3, 117.2 (Ar-C), 21.2 ppm (C$H_3$); MS (EI): $m/z$ (%) 395 (M$^+$+1, 11.2), 394 (M$^+$, 100) [23,24].

### 5.3.17. 3,3′-(1,4-Phenylene)bis(2,6-dimethylquinazolin-4(3H)-one) 6b

Yellow crystals, yield 97%, m.p. 263–265°C; IR (KBr): $v$/cm$^{-1}$ 1676 (C=O), 1610, 1591 (C=N, C=C); $^1$H NMR (400 MHz, DMSO-$d_6$): δ 7.99 (s, 2H, Ar-H), 7.80–7.79 (d, $J$ = 7.6 Hz, 2H, Ar-H), 7.42–7.39 (m, 2H, Ar-H), 7.23 (s, 4H, Ar-H), 2.63 (s, 6H, C$H_3$), 2.26 ppm (s, 6H, C$H_3$); $^{13}$C NMR (100 MHz, DMSO-$d_6$): δ 170.9 (C=O), 144.5 (C=N), 143.1, 138.3, 133.9, 133.0, 128.1, 126.3, 121.9 (Ar-C), 23.6 (C$H_3$), 21.3 ppm (C$H_3$); MS (EI): $m/z$ (%) 423 (M$^+$+1, 6.2), 422 (M$^+$, 100); Anal. Calcd for $C_{26}H_{22}N_4O_2$: C, 73.92; H, 5.25; N, 13.26%; Found: C, 73.96; H, 5.22; N, 13.21%.

### 5.3.18. 3,3′-(1,4-Phenylene)bis(6-chloro-2-methylquinazolin-4(3H)-one) 6c

Yellow crystals, yield 95%, m.p. 287–290°C; IR (KBr): $v$/cm$^{-1}$ 1660 (C = O), 1615, 1597 (C=N, C=C); $^1$H NMR (400 MHz, DMSO-$d_6$): δ 8.09 (s, 2H, Ar-H), 7.55–7.51 (d, $J$ = 7.6 Hz, 2H, Ar-H), 7.46 (s, 4H, Ar-H), 7.12–7.10 (m, 2H, Ar-H), 2.30 ppm (s, 6H, C$H_3$); $^{13}$C NMR (100 MHz, DMSO-$d_6$): δ 170.0 (C=O), 144.7 (C=N), 144.5, 134.2, 133.7, 127.7, 127.2, 122.9, 121.7 (Ar-C), 21.5 ppm (C$H_3$); MS (EI): $m/z$ (%) 464 (M$^+$+2, 62.8), 462 (M$^+$, 100); Anal. Calcd for $C_{24}H_{16}Cl_2N_4O_2$: C, 62.22; H, 3.48; N, 12.09%; Found: C, 62.18; H, 3.51; N, 12.03%.

### 5.3.19. 3,3′-(1,4-Phenylene)bis(6-fluoro-2-methylquinazolin-4(3H)-one) 6d

Yellow crystals, yield 97%, m.p. 269–272°C; IR (KBr): $v$/cm$^{-1}$ 1674 (C=O), 1622, 1599 (C=N, C=C); $^1$H NMR (400 MHz, DMSO-$d_6$): δ 7.98–7.96 (d, $J$ = 7.7 Hz, 2H, Ar-H), 7.66–7.61 (m, 2H, Ar-H), 7.32 (s, 4H, Ar-H), 7.22–7.19 (m, 2H, Ar-H), 2.20 ppm (s, 6H, C$H_3$); $^{13}$C NMR (100 MHz, DMSO-$d_6$): δ 170.2 (C=O), 144.0 (C = N), 162.2, 143.3, 134.0, 125.7, 122.7, 121.2, 120.9, 112.9 (Ar-C), 21.8 ppm (C$H_3$); MS (EI): $m/z$ (%) 431 (M$^+$+1, 6.4), 430 (M$^+$, 100); Anal. Calcd for $C_{24}H_{16}F_2N_4O_2$: C, 66.97; H, 3.75; N, 13.02%; Found: C, 66.95; H, 3.78; N, 12.98%.

### 5.3.20. 3,3′-(1,4-Phenylene)bis(2-methyl-6-nitroquinazolin-4(3H)-one) 6e

Yellow crystals, yield 93%, m.p. 307–310°C; IR (KBr): $v$/cm$^{-1}$ 1683 (C=O), 1614, 1593 (C=N, C=C); $^1$H NMR (400 MHz, DMSO-$d_6$): δ 8.68 (s, 2H, Ar-H), 8.22–8.20 (d, $J$ = 5.5 Hz, 2H, Ar-H), 7.81–7.80 (d, $J$ = 5.5 Hz, 2H, Ar-H), 7.33 (s, 4H, Ar-H), 2.26 ppm (s, 6H, C$H_3$); $^{13}$C NMR (100 MHz, DMSO-$d_6$): δ 170.9 (C=O), 144.2 (C = N), 152.2, 144.7, 134.6, 128.5, 124.2, 121.8, 121.5, 119.5 (Ar-C), 21.6 ppm (C$H_3$); MS (EI): $m/z$ (%) 485 (M$^+$+1, 3.1), 484 (M$^+$, 100); Anal. Calcd for $C_{24}H_{16}N_6O_6$: C, 59.51; H, 3.33; N, 17.35%; Found: C, 59.48; H, 3.39; N, 17.31%.

### 5.3.21. 3,3′-(Sulfonylbis(4,1-phenylene))bis(2-methylquinazolin-4(3H)-one) 7a

Yellow crystals, yield 98%, m.p. 294–297°C; IR (KBr): $v$/cm$^{-1}$ 1689 (C=O), 1615, 1597 (C=N, C=C), 1321 (S=O); $^1$H NMR (400 MHz, DMSO-$d_6$): δ 8.10–8.07 (m, 4H, Ar-H), 7.88–7.83 (m, 4H, Ar-H), 7.58–7.54 (m, 4H, Ar-H), 7.44–7.40 (m, 4H, Ar-H), 2.22 ppm (s, 6H, C$H_3$); $^{13}$C NMR (100 MHz, DMSO-$d_6$): δ 170.2 (C=O), 144.8 (C=N), 145.7, 144.6, 138.5, 134.0, 128.5, 128.1, 126.1, 123.5, 121.3 (Ar-C), 21.9 ppm (C$H_3$); MS (EI): $m/z$ (%) 535 (M$^+$+1, 1.3), 534 (M$^+$, 100); Anal. Calcd for $C_{30}H_{22}N_4O_4S$: C, 67.40; H, 4.15; N, 10.48%; Found: C, 67.44; H, 4.12; N, 10.43%.

### 5.3.22. 3,3′-(Sulfonylbis(4,1-phenylene))bis(2,6-dimethylquinazolin-4(3H)-one) 7b

Light yellow crystals, yield 98%, m.p. 310–313°C; IR (KBr): $v$/cm$^{-1}$ 1680 (C=O), 1622, 1611 (C=N, C=C), 1308 (S=O); $^1$H NMR (400 MHz, DMSO-$d_6$): δ 7.92–7.90 (m, 2H, Ar-H), 7.72 (s, 2H, Ar-H), 7.60–7.52 (m, 2H, Ar-H), 7.42–7.39 (d, $J$ = 7.5 Hz, 4H, Ar-H), 7.306–7.301 (m, 4H, Ar-H), 2.40 (s, 6H, C$H_3$), 2.21 ppm (s, 6H, C$H_3$); $^{13}$C NMR (100 MHz, DMSO-$d_6$): δ 170.5 (C=O), 144.2 (C=N), 144.0, 143.7, 137.5, 133.7, 128.2, 128.0, 127.8, 126.2, 123.1, 121.8 (Ar-C), 23.3 (C$H_3$), 21.6 ppm (C$H_3$); MS (EI): $m/z$ (%) 563 (M$^+$+1, 1.9), 562 (M$^+$, 100); Anal. Calcd for $C_{32}H_{26}N_4O_4S$: C, 68.31; H, 4.66; N, 9.96%; Found: C, 68.27; H, 4.69; N, 9.91%.

### 5.3.23. 3,3′-(Sulfonyl*bis*(4,1-phenylene))*bis*(6-chloro-2-methylquinazolin-4(3*H*)-one) **7c**

Yellow crystals, yield 97%, m.p. 332–335°C; IR (KBr): $\nu$/cm$^{-1}$ 1689 (C=O), 1629, 1607 (C=N, C=C), 1327 (S=O); $^1$H NMR (400 MHz, DMSO-$d_6$): $\delta$ 8.36 (s, 2H, Ar-H), 7.92–7.90 (m, 2H, Ar-H), 7.71–7.62 (m, 2H, Ar-H), 7.47–7.41 (d, $J = 7.8$ Hz, 4H, Ar-H), 7.39–7.32 (m, 4H, Ar-H), 2.29 (s, 6H, C$H_3$); $^{13}$C NMR (100 MHz, DMSO-$d_6$): $\delta$ 171.1 (C=O), 144.0 (C=N), 144.2, 143.9, 138.1, 133.9, 132.8, 128.5, 127.8, 122.8, 121.6 (Ar-C), 21.9 ppm (CH$_3$); MS (EI): $m/z$ (%) 604 (M$^+$+2, 62.9), 602 (M$^+$, 100); Anal. Calcd for C$_{30}$H$_{20}$Cl$_2$N$_4$O$_4$S: C, 59.71; H, 3.34; N, 9.28%; Found: C, 59.74; H, 3.32; N, 9.25%.

### 5.3.24. 3,3′-(Sulfonyl*bis*(4,1-phenylene))*bis*(6-fluoro-2-methylquinazolin-4(3*H*)-one) **7d**

Yellow crystals, yield 93%, m.p. 301–303°C; IR (KBr): $\nu$/cm$^{-1}$ 1692 (C=O), 1621, 1602 (C=N, C=C), 1312 (S=O); $^1$H NMR (400 MHz, DMSO-$d_6$): $\delta$ 7.82–7.80 (m, 2H, Ar-H), 7.69–7.59 (m, 2H, Ar-H), 7.36–7.30 (m, 4H, Ar-H), 7.16–7.15 (m, 2H, Ar-H), 7.14–7.08 (m, 4H, Ar-H), 2.25 (s, 6H, C$H_3$); $^{13}$C NMR (100 MHz, DMSO-$d_6$): $\delta$ 171.1 (C=O), 144.0 (C=N), 161.9, 143.6, 142.7, 138.0, 128.9, 125.6, 122.7, 120.8, 112.9 (Ar-C), 21.8 ppm (CH$_3$); MS (EI): $m/z$ (%) 571 (M$^+$+1, 2.5), 570 (M$^+$, 100); Anal. Calcd for C$_{30}$H$_{20}$F$_2$N$_4$O$_4$S: C, 63.15; H, 3.53; N, 9.82%; Found: C, 63.11; H, 3.57; N, 9.78%.

### 5.3.25. 3,3′-(Sulfonyl*bis*(4,1-phenylene))*bis*(2-methyl-6-nitroquinazolin-4(3*H*)-one) **7e**

Yellow crystals, yield 95%, m.p. 316–319°C; IR (KBr): $\nu$/cm$^{-1}$ 1688 (C=O), 1601, 1589 (C=N, C=C), 1318 (S=O); $^1$H NMR (400 MHz, DMSO-$d_6$): $\delta$ 8.66 (s, 2H, Ar-H), 8.31–8.29 (m, 2H, Ar-H), 8.00–7.98 (m, 4H, Ar-H), 7.66–7.65 (d, $J = 5.5$ Hz, 4H, Ar-H), 7.45–7.40 (m, 2H, Ar-H), 2.21 (s, 6H, C$H_3$); $^{13}$C NMR (100 MHz, DMSO-$d_6$): $\delta$ 171.1 (C=O), 144.2 (C=N), 152.9, 143.2, 137.3, 128.6, 125.0, 122.4, 122.2, 120.6, 118.5 (Ar-C), 21.8 ppm (CH$_3$); MS (EI): $m/z$ (%) 625 (M$^+$+1, 0.9), 624 (M$^+$, 100); Anal. Calcd for C$_{30}$H$_{20}$N$_6$O$_8$S: C, 57.69; H, 3.23; N, 13.46%; Found: C, 57.72; H, 3.20; N, 13.41%.

### 5.3.26. 3-(2-Aminophenyl)-2-methylquinazolin-4(3*H*)-one **8**

White crystals, yield 33%, m.p. 109–111°C; IR (KBr): $\nu$/cm$^{-1}$ 3324–2984 (NH$_2$), 1694 (C=O), 1611, 1594 (C=N, C=C); $^1$H NMR (400 MHz, DMSO-$d_6$): $\delta$ 8.208–8.205 (d, $J = 5.7$ Hz, 1H, Ar-H), 7.93–7.91 (d, $J = 7.8$ Hz, 1H, Ar-H), 7.73–7.70 (d, $J = 5.7$ Hz, 1H, Ar-H), 7.66–7.63 (m, 2H, Ar-H), 7.48–7.44 (m, 2H, Ar-H), 7.28–7.24 (m, 1H, Ar-H), 6.50 (br, 2H, N$H_2$), 2.29 ppm (s, 3H, C$H_3$); $^{13}$C NMR (100 MHz, DMSO-$d_6$): $\delta$ 171.1 (C=O), 154.3 (C=N), 141.7, 137.8, 135.0, 130.9, 130.1, 128.9, 127.7, 127.4, 126.7, 121.9, 117.4, 115.0 (Ar-C), 21.0 ppm (CH$_3$); MS (EI): $m/z$ (%) 252 (M$^+$+1, 100), 251 (M$^+$, 2.6) [23,24].

### 5.3.27. 3,3′-(1,2-Phenylene)*bis*(2-methylquinazolin-4(3*H*)-one) **9**

Yellow crystals, yield 88%, m.p. 187–190°C; IR (KBr): $\nu$/cm$^{-1}$ 1685 (C=O), 1621, 1588 (C=N, C=C); $^1$H NMR (400 MHz, DMSO-$d_6$): $\delta$ 8.11–8.08 (d, $J = 5.7$ Hz, 2H, Ar-H), 7.90–7.88 (d, $J = 7.7$ Hz, 2H, Ar-H), 7.72–7.68 (m, 4H, Ar-H), 7.33–7.30 (d, $J = 9.8$ Hz, 4H, Ar-H), 2.24 ppm (s, 6H, C$H_3$); $^{13}$C NMR (100 MHz, DMSO-$d_6$): $\delta$ 171.7 (C=O), 148.9 (C=N), 142.7, 133.9, 133.7, 127.0, 126.9, 126.6, 123.4, 121.2, 115.7 (Ar-C), 24.1 ppm CH$_3$); MS (EI): $m/z$ (%) 395 (M$^+$+1, 7.9), 394 (M$^+$, 100) [23,24].

### 5.3.28. 3-(3-Aminophenyl)-2-methylquinazolin-4(3*H*)-one **10**

White crystals, yield 33%, m.p. 121–123°C; IR (KBr): $\nu$/cm$^{-1}$ 3365–2990 (NH$_2$), 1687 (C=O), 1609, 1587 (C=N, C=C); $^1$H NMR (400 MHz, DMSO-$d_6$): $\delta$ 8.18–8.15 (d, $J = 7.5$ Hz, 1H, Ar-H), 7.86–7.81 (t, $J = 7.8$ Hz, 1H, Ar-H), 7.76–7.70 (m, 2H, Ar-H), 7.36–7.31 (m, 1H, Ar-H), 6.98–6.94 (m, 2H, Ar-H), 6.78–6.74 (m, 1H, Ar-H), 2.26 ppm (s, 3H, C$H_3$); $^{13}$C NMR (100 MHz, DMSO-$d_6$): $\delta$ 170.4 (C=O), 154.0 (C=N), 145.7, 144.8, 133.2, 129.9, 128.0, 127.5, 126.6, 122.4, 120.0, 118.9, 115.4, 111.6 (Ar-C), 21.4 ppm (CH$_3$); MS (EI): $m/z$ (%) 252 (M$^+$+1, 17.0), 251 (M$^+$, 100) [51].

### 5.3.29. 3,3′-(1,3-Phenylene)*bis*(2-methylquinazolin-4(3*H*)-one) **11**

Light yellow crystals, yield 92%, m.p. 236–238°C; IR (KBr): $\nu$/cm$^{-1}$ 1679 (C=O), 1616, 1607 (C=N, C=C); $^1$H NMR (400 MHz, DMSO-$d_6$): $\delta$ 7.92–7.90 (m, 2H, Ar-H), 7.72 (s, 1H, Ar-H), 7.34–7.28 (m, 4H, Ar-H), 7.10–7.08 (d, $J = 7.5$ Hz, 2H, Ar-H), 7.03–6.99 (m, 3H, Ar-H), 2.30 ppm (s, 6H, C$H_3$); $^{13}$C NMR (100 MHz, DMSO-$d_6$): $\delta$ 171.2 (C=O), 154.3 (C=N), 144.6, 134.4, 133.4, 129.6, 127.8, 127.4,

126.1, 122.4, 121.7, 114.1 (Ar-C), 21.7 ppm (CH$_3$); MS (EI): $m/z$ (%) 395 (M$^+$+1, 22.5), 394 (M$^+$, 100); Anal. Calcd for C$_{24}$H$_{18}$N$_4$O$_2$: C, 73.08; H, 4.60; N, 14.20%; Found: C, 73.11; H, 4.58; N, 14.19%.

## 5.4. In vitro anti-cancer screening

The cytotoxic activities of all newly prepared derivatives were assessed in monolayer cultures by utilizing MTT assay. Two human cancer cell lines (MCF-7 and A549) and one normal cell line (MCF-10A), were maintained in the minimal essential medium (MEM) supplemented with 10% fetal bovine serum (FBS), 2 ml glutamine and 100 units ml$^{-1}$ penicillin in a CO$_2$ incubator in a humidified atmosphere of 5% CO$_2$ and 95% air. The tested derivatives were prepared prior to the experiment by dissolving in 0.1% DMSO and diluted with medium. The cells were then exposed to two concentrations of drugs (10 and 30 μM) in the volume of 100 μM/well. Cells in the control wells received the same volume of medium comprising 0.1% DMSO. After 24 h, the medium was removed and cell cultures were incubated with 100 ml MTT reagent for 5 h at 37°C. The known number of cells was incubated in a 5% CO$_2$ incubator at 37°C in the presence of different concentrations of test compounds. After 48 h of drug incubation, the MTT solution was added in each well and the absorbance was measured by using a microplate reader at 490 nm. The experiment was performed in triplicate. Cell survival was calculated as the percentage of MTT inhibition as % growth inhibition = 100 − (mean OD of individual test group/mean OD of each control group) × 100. The IC$_{50}$ values of the synthesized compounds for two cell lines are summarized in electronic supplementary material, S59, table S3.

Data accessibility. The datasets supporting this article have been uploaded as part of the electronic supplementary material.

Competing interests. I declare I have no competing interests.

Funding. This study was financially supported by the Jouf University, project no. 634/39.

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
