## [Reviewer comments · Royal Society Open Science]

Review History

RSOS-182046.R0 (Original submission)

Review form: Reviewer 1

Is the manuscript scientifically sound in its present form?

Yes

Are the interpretations and conclusions justified by the results?

Yes

Is the language acceptable?

Yes

Is it clear how to access all supporting data?

Yes

Do you have any ethical concerns with this paper?

No

Have you any concerns about statistical analyses in this paper?

No

Recommendation?

Accept with minor revision (please list in comments)

Comments to the Author(s)

The manuscript entitled "Deep eutectic solvent for an expeditious sono-synthesis of novel series of bis-quinazolin-4-one derivatives as potential anticancer agents " by Arafa, describes a green protocol for synthesis of bis-quinazolin-4-one derivatives using deep eutectic solvent under ultrasonic irradiation. The new methodology performs for activating and deactivating starting substrates. The author describes a hypothetical mechanism presented in Scheme 5 that does not agree with me. In the first step the benzoxazine is activated by the components of DEES through the protonation of its carbonyl by tartaric acid. Hydrogen bonding is the main factor that influences the reactivity and selectivity of the process. The reversible hydrogen bonding between tartaric acid and carbonyl groups of benzoxazine make substrate-solvent complex activated. The next condensation of activated carbonyl group with the amine in the DES leads to the formation of a cationic intermediate with the following loss of a water molecule (Di Gioia et al. *Molecules* 2018, 23, 1891). Council to the author to completely review the hypothetical mechanism before publishing it, reporting the correct corrections with the bibliographic reference (Di Gioia et al. *Molecules* 2018, 23, 1891). The reference 21 is not relevant to the hypothesised reaction mechanism.

Review form: Reviewer 2

Is the manuscript scientifically sound in its present form?

No

Are the interpretations and conclusions justified by the results?

No

Is the language acceptable?

Yes

Is it clear how to access all supporting data?

Not Applicable

Do you have any ethical concerns with this paper?

No

Have you any concerns about statistical analyses in this paper?

I do not feel qualified to assess the statistics

Recommendation?

Major revision is needed (please make suggestions in comments)

Comments to the Author(s)

See attached file (Appendix A).

Review form: Reviewer 3

Is the manuscript scientifically sound in its present form?

Yes

Are the interpretations and conclusions justified by the results?

Yes

Is the language acceptable?

No

Is it clear how to access all supporting data?

Yes

Do you have any ethical concerns with this paper?

No

Have you any concerns about statistical analyses in this paper?

No

Recommendation?

Accept with minor revision (please list in comments)

Comments to the Author(s)

Arafa report the condensation of diamines with acetantranil with the main novelty being the use of a "deep eutectic solvent." Admittedly, The key condensation reaction between diamines and acetantranils has been known since as early as 1911, this work was not cited and should be, considering not only it's key precedent, but many of the same molecules are prepared in this work. Please note that the reference below does have the wrong structure for acetantranils, an error likely due to the lack of spectroscopic methods from this time (early 1900's).

Journal of the American Chemical Society (1911), 33, 949-62

This age-old reaction appears to work well in DESs, and they author reports an improved scope. The authors also report cytotoxicity studies. This reviewer is not qualified to comment on this data (interpretation or accuracy). I see a table of data, but no "raw" or "supporting" data. Is this common? This review is unsure.

Address the following before further considering this work:

1. The citation referenced above
2. Syntax. This is a big problem throughout the manuscript.
3. The mechanism – (a) the DES structure seems suspect: the chloride is binding to the carboxylates only on the tartaric acid. Is this likely the case? Shown is a 9-membered chelate. I would suspect that tartrate chelates to form smaller rings? Is there evidence to back this structure up? Is this just a simplification of the DES? (b) an imine with only one long pair is shown having H-bonds to two protons. This is impossible...(c) L-(+)-Tartaric acid's stereochemistry is omitted.
4. This reviewer does not understand the temperature column in Table 1. (i.e. 50/US) I think it might be an abbreviation for Ultrasound. Why is this in the temperature column if this is the case? It is confusing.
5. The SI – There are only 1H NMRs shown? If this is to the journals standard then OK. Include 13C spectra if journal requires this.

6. The SI - The ^1H NMR spectra do not have integrations. I also highly recommend including a structure of the molecule on the spectrum. This will facilitate data interpretation and confirm purity.

Decision letter (RSOS-182046.R0)

03-Jan-2019

Dear Professor Arafa:

Title: Deep eutectic solvent for an expeditious sono-synthesis of novel series of bis-quinazolin-4-one derivatives as potential anticancer agents
Manuscript ID: RSOS-182046

The editor assigned to your manuscript has now received comments from reviewers. We would like you to revise your paper in accordance with the referee and Subject Editor suggestions which can be found below (not including confidential reports to the Editor). Please note this decision does not guarantee eventual acceptance.

Please submit your revised paper before 26-Jan-2019. Please note that the revision deadline will expire at 00.00am on this date. If we do not hear from you within this time then it will be assumed that the paper has been withdrawn. In exceptional circumstances, extensions may be possible if agreed with the Editorial Office in advance. We do not allow multiple rounds of revision so we urge you to make every effort to fully address all of the comments at this stage. If deemed necessary by the Editors, your manuscript will be sent back to one or more of the original reviewers for assessment. If the original reviewers are not available we may invite new reviewers.

On behalf of the Subject Editor Professor Anthony Stace and the Associate Editor Dr Andrew Harned.

RSC Associate Editor:

Comments to the Author:

There appears to be some enthusiasm for this work by the reviewers, however they feel the authors have missed the mark a bit with regard to presentation and scientific rigor. The authors should revise their manuscript by carefully considering the comments and concerns raised by the reviewers, especially those related to the proposed mechanism and biological activity.

In addition to those comments, I have a couple of suggestions:

(1) Table 3 does not add to the discussion as the same information can be found in Figure 3. Please move Table 3 to the supporting information.

(2) Please add structures to the ¹H NMR spectra in the supporting information and add copies of the ¹³C NMR spectra.

RSC Subject Editor:

Comments to the Author:

(There are no comments.)

Reviewers' Comments to Author:

Reviewer: 1

Comments to the Author(s)

The manuscript entitled "Deep eutectic solvent for an expeditious sono-synthesis of novel series of bis-quinazolin-4-one derivatives as potential anticancer agents" by Arafa, describes a green protocol for synthesis of bis-quinazolin-4-one derivatives using deep eutectic solvent under ultrasonic irradiation. The new methodology performs for activating and deactivating starting substrates. The author describes a hypothetical mechanism presented in Scheme 5 that does not agree with me. In the first step the benzoxazine is activated by the components of DEES through the protonation of its carbonyl by tartaric acid. Hydrogen bonding is the main factor that influences the reactivity and selectivity of the process. The reversible hydrogen bonding between tartaric acid and carbonyl groups of benzoxazine make substrate-solvent complex activated. The next condensation of activated carbonyl group with the amine in the DES leads to the formation of a cationic intermediate with the following loss of a water molecule (Di Gioia et al. *Molecules* 2018, 23, 1891). Council to the author to completely review the hypothetical mechanism before publishing it, reporting the correct corrections with the bibliographic reference (Di Gioia et al. *Molecules* 2018, 23, 1891). The reference 21 is not relevant to the hypothesised reaction mechanism.

Reviewer: 2

Comments to the Author(s)

See attached file

Reviewer: 3

Comments to the Author(s)

Arafa report the condensation of diamines with acetantranil with the main novelty being the use of a "deep eutectic solvent." Admittedly, The key condensation reaction between diamines and acetantranils has been known since as early as 1911, this work was not cited and should be, considering not only it's key precedent, but many of the same molecules are prepared in this work. Please note that the reference below does have the wrong structure for acetantranils, an error likely due to the lack of spectroscopic methods from this time (early 1900's).

Journal of the American Chemical Society (1911), 33, 949-62

This age-old reaction appears to work well in DESs, and they author reports an improved scope. The authors also report cytotoxicity studies. This reviewer is not qualified to comment on this data (interpretation or accuracy). I see a table of data, but no "raw" or "supporting" data. Is this common? This review is unsure.

Address the following before further considering this work:

1. The citation referenced above
2. Syntax. This is a big problem throughout the manuscript.
3. The mechanism - (a) the DES structure seems suspect: the chloride is binding to the carboxylates only on the tartaric acid. Is this likely the case? Shown is a 9-membered chelate. I would suspect that tartrate chelates to form smaller rings? Is there evidence to back this structure up? Is this just a simplification of the DES? (b) an imine with only one long pair is shown having H-bonds to two protons. This is impossible...(c) L-(+)-Tartaric acid's stereochemistry is omitted.
4. This reviewer does not understand the temperature column in Table 1. (i.e. 50/US) I think it might be an abbreviation for Ultrasound. Why is this in the temperature column if this is the case? It is confusing.
5. The SI - There are only 1H NMRs shown? If this is to the journals standard then OK. Include 13C spectra if journal requires this.
6. The SI - The 1H NMR spectra do not have integrations. I also highly recommend including a structure of the molecule on the spectrum. This will facilitate data interpretation and confirm purity.

Author's Response to Decision Letter for (RSOS-182046.R0)

See Appendix B.

RSOS-182046.R1 (Revision)

Review form: Reviewer 1

Is the manuscript scientifically sound in its present form?

Yes

Are the interpretations and conclusions justified by the results?

Yes

Is the language acceptable?

Yes

Is it clear how to access all supporting data?

Yes

Do you have any ethical concerns with this paper?

No

Have you any concerns about statistical analyses in this paper?

No

Recommendation?

Accept as is

Comments to the Author(s)

I recommend the publication of the manuscript as it is

Review form: Reviewer 2

Is the manuscript scientifically sound in its present form?

Yes

Are the interpretations and conclusions justified by the results?

Yes

Is the language acceptable?

Yes

Is it clear how to access all supporting data?

Yes

Do you have any ethical concerns with this paper?

No

Have you any concerns about statistical analyses in this paper?

No

Recommendation?

Accept with minor revision (please list in comments)

Comments to the Author(s)

The author made all corrections/changes I suggested and this version seems to be better formatted and correct. The syntax problems also seemed to be solved. In my opinion, Figure 3 must show the values for cytotoxic activity against MCF-7 and A549 cell lines. Page 6, lines 36 and 37 "The obtained results conclude..."; please correct this sentence as "the results cannot conclude anything".

Decision letter (RSOS-182046.R1)

06-Feb-2019

Dear Professor Arafa:

Title: Deep eutectic solvent for an expeditious sono-synthesis of novel series of bis-quinazolin-4-one derivatives as potential anticancer agents
Manuscript ID: RSOS-182046.R1

Thank you for submitting the above manuscript to Royal Society Open Science. On behalf of the Editors and the Royal Society of Chemistry, I am pleased to inform you that your manuscript will be accepted for publication in Royal Society Open Science subject to minor revision in accordance with the referee suggestions. Please find the reviewers' comments at the end of this email.

The reviewers and handling editors have recommended publication, but also suggest some minor revisions to your manuscript. Therefore, I invite you to respond to the comments and revise your manuscript.

Because the schedule for publication is very tight, it is a condition of publication that you submit the revised version of your manuscript before 15-Feb-2019. Please note that the revision deadline will expire at 00.00am on this date. If you do not think you will be able to meet this date please let me know immediately.

1) A text file of the manuscript (tex, txt, rtf, docx or doc), references, tables (including captions) and figure captions. Do not upload a PDF as your "Main Document".

- 2) A separate electronic file of each figure (EPS or print-quality PDF preferred (either format should be produced directly from original creation package), or original software format)
- 3) Included a 100 word media summary of your paper when requested at submission. Please ensure you have entered correct contact details (email, institution and telephone) in your user account
- 4) Included the raw data to support the claims made in your paper. You can either include your data as electronic supplementary material or upload to a repository and include the relevant doi within your manuscript
- 5) All supplementary materials accompanying an accepted article will be treated as in their final form. Note that the Royal Society will neither edit nor typeset supplementary material and it will be hosted as provided. Please ensure that the supplementary material includes the paper details where possible (authors, article title, journal name).

Best wishes,

Dr Laura Smith
Publishing Editor, Journals

On behalf of the Subject Editor Professor Anthony Stace and the Associate Editor Dr Andrew Harned.

RSC Associate Editor:

Comments to the Author:

The reviewers are generally pleased with the changes made to the previous manuscript. One reviewer has asked that one sentence be modified. I agree with this suggestion, as the results cannot conclude anything. The authors, however, are able to reach a conclusion. The same reviewer also appears to be asking for actual IC50 values to be added to Figure 3. I don't think this is necessary as the current version conveys the comparative data quite nicely and the actual values are available in the supporting information. There is even a note in the main text directing the reader to the SI for this information.

I will gladly accept a revised manuscript if the authors simply modify the offending sentence.

RSC Subject Editor:
Comments to the Author:
(There are no comments.)

Reviewer comments to Author:
Reviewer: 1

Comments to the Author(s)
I recommend the publication of the manuscript as it is

Reviewer: 2

Comments to the Author(s)
The author made all corrections/changes I suggested and this version seems to be better formatted and correct. The syntax problems also seemed to be solved. In my opinion, Figure 3 must show the values for cytotoxic activity against MCF-7 and A549 cell lines. Page 6, lines 36 and 37 "The obtained results conclude..."; please correct this sentence as "the results cannot conclude anything".

Author's Response to Decision Letter for (RSOS-182046.R1)

See Appendix C.

Decision letter (RSOS-182046.R2)

15-Feb-2019

Dear Professor Arafa:

Title: Deep eutectic solvent for an expeditious sono-synthesis of novel series of bis-quinazolin-4-one derivatives as potential anticancer agents
Manuscript ID: RSOS-182046.R2

It is a pleasure to accept your manuscript in its current form for publication in Royal Society Open Science. The chemistry content of Royal Society Open Science is published in collaboration with the Royal Society of Chemistry.

On behalf of the Subject Editor Professor Anthony Stace and the Associate Editor Dr Andrew Harned.

RSC Associate Editor
Comments to the Author:
(There are no comments.)

Reviewer(s)' Comments to Author:

Appendix A

Deep eutectic solvent for an expeditious sono-synthesis of novel series of bis-quinazolin-4-one derivatives as potential anticancer agents

Arafa, W.

RSOS-182046

The article describes a study on the synthesis of bis-quinazolin-4-ones using a sonochemical approach in deep eutectic mixtures as solvent/catalyst. Although the reported results can be considered as a good contribution to the area, the author failed to furnish enough and accurate evidence to support his contribution. In addition, I recommend a careful English review of the text and I suggest the author to look into consideration the following main points.

1. Page 1, lines 44 and 59 (and others); "utilize" cannot be used as a noun and it must be changed by "the use" or "the utilization".
2. Page 1, line 28: what is a benzoheterocycle?
3. The use of DES as solvent/catalyst for the synthesis of heterocycles is the subject of rapidly expanding literature base and should be exploited more deeply in the Introduction.
4. Taking into account the optimization study described in page 2 and Table 1, I would expect reactions conducted under ultrasonic irradiation using H₂O at high temperatures. Compare entry 8 with entries 15 & 16: is the reaction temperature an important variable? **I strongly recommend the author to perform a control experiment using H₂O at 90°C (or even at 100°C) under US, in order to verify the actual role of temperature and the deep eutectic mixture on the reaction efficiency.**
5. Please indicate if yields shown in Table 2 are isolated or conversion.
6. Page 4, lines 30,31: not all reactions were quantitative as some described yields are not >99%. In addition purification by chromatography was not necessary but the products were purified by recrystallization. The efficiency of the method is over estimated in that sentence.
7. Page 3, lines 51-53: "*Owing to the fact that heterocycles bearing halogens are significant building blocks in the assembly of plentiful of pharmaceuticals and natural products...*" this sentence must be reviewed because its meaning is quite broad.
8. Page 4, lines 37-39: "*Finally, the required product was formed through dehydration. From the proposed mechanism, **it obvious concluded** that, L-(+)-tartaric acid choline chloride performs a dual function; solvent and catalyst.*" I do not believe this conclusion is obvious. The author has not provided any mechanistic evidence (despite of any logical suggestions) for the study of the mechanism of such transformation.
9. Mechanism depicted in Scheme 5 (page 5): I think there is a misrepresentation of the H-bond activation of benzo[d][1,3]oxazin-4-one (**1**) by DES. In fact, the N atom in **1** cannot be involved in two H-bonds, as represented (just one pair of electrons). In addition, several curved arrows are missing, when nucleophilic attacks were represented.
10. Concerning the MTT test, it is important to mention that this test reveals cytotoxicity profiles, The "anticancer activity" term used throughout the text must be corrected.

11. The cytotoxicity activity was evaluated in normal cells? It is important to check the selectivity of the title compounds, using normal cell lines such as HUVEC (endothelial cells), human mammary epithelial cell (MRF10a), among many others. In this scenario, I was intrigued by the fact that the most active compound is a nitro-derivative (**7e**). Once biological active nitro-compounds are recognized as highly toxic, I speculate if **7e** would also exhibit high cytotoxic activity against normal cell lines. **I strongly recommend the author to run in vitro cytotoxicity evaluation against normal cell lines for the most active compounds.**
12. Page 8, lines 52-54: halogen atoms and NO₂ group are not "electron deficient groups". They are electron withdrawing groups, modifying the electron density of the aromatic ring. Similarly, the methyl group is not "electron rich"; in fact the electron donating ability of alkyl groups is somewhat limited. I suggest this argumentation should be entirely reviewed.
13. Finally, in the Conclusion section, the author overestimated the "potential anticancer activity" of the studied compounds in the article. Once again, the in vitro evaluation of cytotoxic activity is only the first step in the development of an anticancer drug. I think it is not appropriate to use "anticancer activity" (in vivo) as synonym for "cytotoxic activity" (in vitro).

Appendix B

A point-by-point response to the comments made by the RSC Associate Editor and Referees

Entry	Reviewer	Comments	Responses
1.	RSC Associate Editor	Table 3 does not add to the discussion as the same information can be found in Figure 3. Please move Table 3 to the supporting information.	Done
2.		Please add structures to the ^1H NMR spectra in the supporting information and add copies of the ^{13}C NMR spectra.	Done
3.	Reviewer: 1	The author describes a hypothetical mechanism presented in Scheme 5 that does not agree with me. Council to the author to completely review the hypothetical mechanism before publishing it, reporting the correct corrections with the bibliographic reference (Di Gioia et al. Molecules 2018, 23, 1891).	The proposed mechanism has been changed according to the recommended reference (Di Gioia et al. Molecules , 2018, 23 , 1891).
4.		The reference 21 is not relevant to the hypothesised reaction mechanism.	Reference No. 22 was omitted and the recommended one was cited: Di Gioia et al., Molecules , 2018, 23 , 1891.
5.	Reviewer: 2	Page 1, lines 44 and 59 (and others); "utilize" cannot be used as a noun and it must be changed by "the use" or "the utilization".	Done.
6.		Page 1, line 28: what is a benzoheterocycle?	The word "benzoheterocycle" was replaced by "nitrogen-containing heterocycles". (highlighted in yellow color). However, heterocyclic compounds that fused with benzene ring well known as "benzoheterocycles", the same word was mentioned in many reports such as: Chem. Commun. , 2018, 54 , 12602.
7.		The use of DES as solvent/catalyst for the synthesis of heterocycles is the subject of rapidly expanding literature base and should be exploited more deeply in the	In the introduction part, the paragraph relating to DES has been modified (highlighted in yellow color).

		Introduction.	
8.		Taking into account the optimization study described in page 2 and Table 1, I would expect reactions conducted under ultrasonic irradiation using H ₂ O at high temperatures. Compare entry 8 with entries 15 & 16: is the reaction temperature an important variable? I strongly recommend the author to perform a control experiment using H ₂ O at 90 °C (or even at 100 °C) under US, in order to verify the actual role of temperature and the deep eutectic mixture on the reaction efficiency.	A controlled experiment (Table 1, entry 8) using H ₂ O at 95 °C was performed and the desired product was obtained in 77% (in 35 min). This result was mentioned in the discussion part (highlighted in yellow). This clarifies the important role of DES "as mentioned in discussion part".
9.		Please indicate if yields shown in Table 2 are isolated or conversion.	 Isolated yields. The phrase: "Isolated yields" has been mentioned below Table 2.
10		Page 4, lines 30,31: not all reactions were quantitative as some described yields are not >99%. In addition purification by chromatography was not necessary but the products were purified by recrystallization. The efficiency of the method is overestimated in that sentence.	The sentence " As the reactions were quantitative, no column purification was necessary in all cases. " has been omitted.
11		Page 3, lines 51-53: "Owing to the fact that heterocycles bearing halogens are significant building blocks in the assembly of plentiful of pharmaceuticals and natural products..." this sentence must be reviewed because its meaning is quite broad.	Reference No. 19 was cited. (highlighted in yellow color).
12		Page 4, lines 37-39: "Finally, the required product was formed through dehydration. From the proposed mechanism, it is obvious that, L-(+)-tartaric acid choline chloride performs a dual function; solvent and catalyst." I do not believe this conclusion is obvious. The author has not provided any mechanistic evidence (despite of any logical suggestions) for the study of the mechanism of such	Reference No. 22 has been cited that supporting the suggested mechanism.

		transformation.	
13		Mechanism depicted in Scheme 5 (page 5): I think there is a misrepresentation of the H-bond activation of benzo[d][1,3]oxazin-4-one (1) by DES. In fact, the N atom in 1 cannot be involved in two H-bonds, as represented (just one pair of electrons). In addition, several curved arrows are missing, when nucleophilic attacks were represented.	The proposed mechanism has been changed according to reference: Di Gioia et al. Molecules , 2018, 23 , 1891. Also, the missing curved arrows have been drawn.
14		Concerning the MTT test, it is important to mention that this test reveals cytotoxicity profiles, The “anticancer activity” term used throughout the text must be corrected.	Done.
15		The cytotoxicity activity was evaluated in normal cells? It is important to check the selectivity of the title compounds, using normal cell lines such as HUVEC (endothelial cells), human mammary epithelial cell (MRF10a), among many others. In this scenario, I was intrigued by the fact that the most active compound is a nitro-derivative (7e). Once biological active nitro-compounds are recognized as highly toxic, I speculate if 7e would also exhibit high cytotoxic activity against normal cell lines. I strongly recommend the author to run in vitro cytotoxicity evaluation against normal cell lines for the most active compounds.	The cytotoxicity activity, for the most active products, was evaluated in normal breast cell line (MCF-10A) and the results were mentioned in the discussion part (highlighted in yellow color).
16		Page 8, lines 52-54: halogen atoms and NO ₂ group are not “electron deficient groups”. They are electron withdrawing groups, modifying the electron density of the aromatic ring. Similarly, the methyl group is not “electron rich”; in fact the electron donating ability of alkyl groups is somewhat limited. I suggest this argumentation should be entirely reviewed.	Done. Both “electron deficient” and “electron rich” have been replaced by “electron withdrawing” and “electron donating”, respectively.

17		Finally, in the Conclusion section, the author overestimated the “potential anticancer activity” of the studied compounds in the article. Once again, the in vitro evaluation of cytotoxic activity is only the first step in the development of an anticancer drug. I think it is not appropriate to use “anticancer activity” (in vivo) as synonym for “cytotoxic activity” (in vitro).	Done.
18	Reviewer: 3	The citation referenced above	Done. “within reference No. 4” (highlighted in yellow color).
19		Syntax. This is a big problem throughout the manuscript.	Done.
20		The mechanism – (a) the DES structure seems suspect: the chloride is binding to the carboxylates only on the tartaric acid. Is this likely the case? Shown is a 9-membered chelate. I would suspect that tartrate chelates to form smaller rings? Is there evidence to back this structure up? Is this just a simplification of the DES? (b) an imine with only one long pair is shown having H-bonds to two protons. This is impossible... (c) L-(+)-Tartaric acid's stereochemistry is omitted.	a) The chloride is now binding to both the carboxylate on the tartaric acid and choline moiety. b) The proposed mechanism has been changed. c) L-(+)-Tartaric acid isomer was used in the preparation of DES while, other stereoisomers did not. For simplification, L-(+)-Tartaric acid was drawn in Chart 5 without regarding to its stereochemistry.
21		This reviewer does not understand the temperature column in Table 1. (i.e. 50/US) I think it might be an abbreviation for Ultrasound. Why is this in the temperature column if this is the case? It is confusing.	 An abbreviation for Ultrasound has been mentioned along with Table 1. (highlighted in yellow color). To the column of temperature, word “Method” has been added.
22		The SI – There are only ^1H NMRs shown? If this is to the journals standard then OK. Include ^{13}C spectra if journal requires this.	Done, copies of ^{13}C NMR spectra have been attached with SI file.
23		The SI – The ^1H NMR spectra do not have integrations. I also highly recommend including a structure of the molecule on the spectrum. This will facilitate data	 All details about integrations were mentioned in details in experimental part. All chemical structures have

		interpretation and confirm purity.	been attached within the spectra.
--	--	------------------------------------	-----------------------------------

Appendix C

The response to the comment made by the Referee

Entry	Comment	Responses
1.	Page 6, lines 36 and 37 "The obtained results conclude..."; please correct this sentence as "the results cannot conclude anything".	Corrected: From the obtained results, we can conclude that these derivatives might be further used as promising anticancer agents.